# Preparation and Application of Fluorine-Free Finishing Agent with Excellent Water Repellency for Cotton Fabric

**DOI:** 10.3390/polym13172980

**Published:** 2021-09-02

**Authors:** Chengbing Yu, Kaiqin Shi, Jinyan Ning, Zhe Zheng, Hualong Yu, Zhenxuan Yang, Jun Liu

**Affiliations:** 1School of Materials Science and Engineering, Shanghai University, Shanghai 201800, China; skqin112@shu.edu.cn (K.S.); icepoorzz@shu.edu.cn (Z.Z.); yhl2018@shu.edu.cn (H.Y.); kazzaz@shu.edu.cn (Z.Y.); 2Materials Genome Institute, Shanghai University, Shanghai 200444, China; jyning@t.shu.edu.cn; 3Shanghai Institute of Quality Inspection and Technical Research, Shanghai 201114, China

**Keywords:** polysiloxane, fluorine-free, organic–inorganic hybrid, water repellency, cotton fabric

## Abstract

Water repellent is an important functional finish for cotton fabric. However, cotton fabrics often have poor washing resistance and other performances after actual finishing. In this study, based on the structural characteristics of cotton fiber and durability of water repellent, a cross-linked amino long-chain alkyl polysiloxane (CAHPS) was first prepared, and then reacted with modified silica. Finally, a chemically bonded organic–inorganic nanohybrid cross-linked polysiloxane (rSiO_2_–CAHPS) was fabricated. Furthermore, the rSiO_2_–CAHPS was emulsified to obtain a durable fluorine-free water repellent. The water repellent finishing for cotton fabric was carried out by the pad–dry–cure process. After finishing, the cotton fabric had good resistance to conventional liquids and excellent washing resistance, and still maintained good water repellency after 30 rounds of soaping. Moreover, properties including air permeability, mechanical property and whiteness are hardly affected after finishing. SEM and XPS characterization show that a layer of dense silicon film is formed on the surface of cotton fabric by rSiO_2_–CAHPS water repellent. The existence of nanosilica can improve the surface roughness of cotton fibers. The synergistic effect of fiber matrix, nanoparticles and CAHPS endows the fabric with a micro/nano-multi-scale micro-rough structure, which improves the water repellency of cotton fabric after water repellent finishing.

## 1. Introduction

As important natural fiber textiles, cotton fiber textiles are favored by consumers for their softness, moisture absorption, air permeability, skin friendliness, warmth and other advantages, but they also have some problems, such as not being easy to dry after absorbing moisture and easily sticking to the skin, causing discomfort and breeding bacteria. Therefore, it is necessary to conduct water repellent finishing for cotton fabric [1,2]. This can broaden the scope of cotton fabric use, improve its economic value, meet the growing market demand for functional textiles and make it have broad application prospects [3,4].

The water repellent finishing of cotton fabric is generally achieved by physical or chemical methods to reduce the surface energy and change the roughness of the fabric surface. The commonly used finishing methods include laminating, coating, dipping and rolling, etc., and commonly used finishing agents are fluorine and silicon finishing agents [5,6,7,8]. Although the organic fluorine water repellents containing C8 structure (-C_8_F_17_) feature excellent water repellent properties, they contain perfluorooctane sulfonate (PFOS) and perfluorooctanoic acid (PFOA), which are difficult to degrade and have high toxicity and bioaccumulation [9,10]. For these reasons, they have been restricted in use, thus the development of fluorine-free water repellents is of great importance [11,12,13].

Silicone organic compounds, with advantages of good biocompatibility and low price of raw materials, can form a water repellent interface with low surface tension on the surface of cotton fibers. The fabric presents good water resistance performance and good hand feel after finishing, which has attracted wide attention from researchers and users [14,15,16]. As for polysiloxane water repellent agents, fabric finished with polydimethylsiloxane can display good water repellent effects, but the washing resistance is not good enough. When hydrogen-containing polysiloxane is used as a water repellent agent, the silicon hydrogen bond on the fabric is easy to hydrolyze under the effect of a catalyst due to its high activity. After hydrolysis of the silicon hydrogen bond, it dehydrates and condenses to produce cross-linking, which endows the fabric with a durable water repellent effect, but this results in hard hand feel of the fabric. When it is used together with polydimethylsiloxane as a water repellent finishing agent to finish the fabric, not only is the water repellency marvelous, but the film forming property and hand feel are also improved [17]. In addition, hydroxyl-modified polysiloxanes, amino polysiloxanes and amino-modified fluorinated polysiloxanes are also commonly used as water repellents to endow fabrics with more durable water repellency [18].

To further improve the performance of silicone water repellent, researchers introduced other types of chain segments into the molecular chain and further combined them with different types of materials by cross-linking, grafting or hydrolysis and condensation of siloxane, which improved the film-forming property, hydrophobicity, long-term resistance, high and low temperature resistance, water resistance and mechanical strength of the polymer. There have been reports in the literature about the modification of silicone water repellent with polyurethane or acrylate [19,20,21]. Among the existing water repellent finishing methods, long-chain alkane compounds can provide cotton fabric with good water repellent properties, but these long-chain alkane compounds lack groups that can react with cotton fibers, resulting in poor washing resistance and severely affecting the hand feel and air permeability [22,23].

It is well known that the combination micro/nanostructures with hierarchical structures and low surface energy materials is an effective way to construct water repellent surfaces [5,6]. Nanoparticles play a significant role in changing the surface roughness of fabrics and can effectively improve hydrophobic properties [7,8]. Nanosilica is commonly used for the surface modification. The nanoparticles modified by silicone do not damage the transparency of the product, but also effectively improve the surface hydrophobicity of cotton fabric [24,25]. However, the agglomeration effect and incompatibility of nanosilica will prevent them from mixing with the medium to form a stable system. Meanwhile, the micro- and nanostructures formed by physical deposition are easily worn off by mechanical force, causing the loss of hydrophobic properties [26]. Additionally, although many techniques for constructing bionic rough surfaces have been developed, such as sol-gel [27,28], etching [29,30], self-assembly [31,32], deposition [33,34,35] and solution impregnation method [36,37], these methods often require harsh preparation conditions, special processing equipment and tedious operational steps. As a result, they are not suitable for industrialization or application in the water repellent finishing of cotton fabric [20,21].

In this study, based on the structural characteristics of cotton fiber, a cross-linked amino long-chain alkyl polysiloxane (CAHPS) was first synthesized by alkaline equilibrium polymerization, and then it underwent a ring-opening addition reaction with modified nanosilica to prepare nanohybrid cross-linked polysiloxanes (rSiO_2_–CAHPS) [15]. The structures of the products were characterized, and the application properties of the finished cotton fabric were further tested and analyzed. The water repellency, resistance to conventional liquids, washing resistance, abrasion resistance, air permeability, fracture strength and whiteness of the finished cotton fabric were tested, and the micromorphology and chemical composition of the finished cotton fabric surface were analyzed by SEM and XPS. Finally, we studied and proposed the film formation mechanism of the water repellency on the fiber surface.

## 2. Materials and Methods

### 2.1. Materials

Octamethyl cyclotetrasiloxane (D_4_) was supplied by Osbang New Material Co., Ltd. (Shenzhen, China). γ-aminopropylmethyl diethoxysilane (APDES) was obtained from Chuang Chemical Aid Co., Ltd. (Nanjing, China). Hexadecyltrimethoxysilane (HDTMS) and 3-Glycidoxypropyltrimethoxysilane (GPTMS) were both supplied by Youpu Chemical Co., Ltd. (Nanjing, China). Tetratrimethyl ammonium hydroxide (TMAH), isopropanol, methylbenzene, anhydrous ethanol, glacial acetic acid and hydrochloric acid were all purchased from Sinopharm Chemical Reagents Co., Ltd. (Shanghai, China). All chemicals were used as received. Three kinds of silica with sizes of 20 nm, 100 nm and 1 μm were provided by Emperor Metal Materials Co., Ltd. (Xingtai, China). Isomeric alcohol ethoxylates 1305 and 1307 were offered by Jingxin Chemical Technology Co., Ltd. (Guangzhou, China). Water repellent WP-107A was kindly supplied by Longyang Dyeing & Finishing Co., Ltd. (Nangtong, China). Deionized water was used throughout the study. The knitted fabrics were a double interlock knit structure at 40/1s combed compact knitting, which were kindly supplied by New Silk Knitted Garment Company (Nantong, China).

### 2.2. Preparation Procedures

#### 2.2.1. Synthesis of Polysiloxane

One hundred grams of D_4_, 8 g APDES and 6 g HDTMS were added to a three-necked flask equipped with a stirrer, thermometer and reflux condenser tube. The mixture was stirred evenly and heated to 90 °C. TMAH was then added to react for 0.5 h. The temperature was increased to 110 °C and maintained for 6 h. At the end of the reaction, the system was heated to 140 °C for 0.5 h to decompose TMAH. The low boiling material was removed by vacuum distillation for 0.5 h. After cooling to room temperature, a colorless, clear and viscous liquid was finally obtained, namely CAHPS (Scheme 1a). ^1^H NMR (400 MHz, CDCl_3_, *δ*, ppm): 7.42 (s, 4H, phenyl ring); 0.11 (Si-CH_3_, aH), 0.53 (Si-CH_2_, bH), 1.54 (CH_2_, cH), 2.67 (CH_2_, dH), 1.20 (CH_2_-NH_2_, eH), 1.25 (-CH_2_-, fH), 0.89 (CH_2_-CH_3_, gH).

#### 2.2.2. Modification of Silica

First, 0.6 g silica and 20 mL toluene were added to a beaker and sonicated for 30 min at room temperature to obtain a homogeneous suspension. Then, 0.2 mol/L GPTMS/toluene solution was added dropwise to the above suspension, maintaining ultrasound for 5 min and transferring it to a three-necked flask equipped with a reflux condenser and thermometer. The reaction was conducted under magnetic stirring for 6 h at a temperature of 90 °C. The reaction solution was centrifuged to obtain GPTMS-modified silica. The modified silica was ultrasonically dispersed with anhydrous ethanol, centrifuged three times and then dried in a vacuum oven at 60 °C for 8 h. The white powder of silica modified by GPTMS was obtained and recorded as GPTMS-SiO_2_ (Scheme 1b).

#### 2.2.3. Synthesis of rSiO_2_–CAHPS

One hundred grams of CAHPS, 15 g GPTMS-SiO_2_ and 50 mL isopropanol were added to a 250 mL three-necked flask and heated to 52 °C. The reaction was carried out under stirring for 4 h. A translucent, viscous liquid was produced by vacuum distillation, which was recorded as rSiO_2_–CAHPS (Scheme 1c).

#### 2.2.4. Preparation of Fluorine-Free Water Repellent Agent

Twenty grams of rSiO_2_–CAHPS and a compound emulsifier consisting of 0.48 g isomeric alcohol ethoxylates 1305 and 0.12 g isomeric alcohol ethoxylates 1307 were added to a three-neck flask. After mixing them thoroughly by stirring, deionized water was slowly added, followed by violent stirring for 15 min. The pH value was adjusted to 5–6 with acetic acid. A milky white liquid was obtained, which was durable fluorine-free water repellent (Scheme 1d).

### 2.3. Finishing with rSiO_2_–CAHPS Water Repellent

The fabric was immersed in a finishing solution (20–120 g/L) at room temperature for 30 s. It was then passed through a vertical two-roll padder (Weida Machinery Co., Ltd., Shaoxing, China) to obtain 65–90% pick-up using a two-dip two-nip procedure. The treated fabrics were dried at 100 °C for 3 min, and then cured at 150–170 °C for 60–180 s to obtain the treated cotton fabrics.

### 2.4. Characterization Methods

Fourier transform infrared spectra (FT-IR) were recorded on a Nicolet 380 infrared spectrophotometer (Thermo Fisher Scientific, Waltham, MA, USA) with a disc of KBr from 4000 cm^−1^ to 400 cm^−1^. Nuclear magnetic resonance (^1^H NMR) spectra using deuterated chloroform (CDCl_3_) as a solvent and tetramethylsilane (TMS) as an internal standard were recorded with a Bruker Avance 500 spectrometer (Bruker, Fällanden, Switzerland) under room temperature.

CAHPS and rSiO_2_–CAHPS were put in a crucible and tested using Q550 thermal weight loss analyzer (TA Instruments, New Castle, DE, USA). The samples were heated from room temperature to 750 °C in a N_2_ atmosphere, and the weight change of the samples during the heating process was measured at a heating rate of 15 °C/min.

The water contact angle (WCA) of the cotton fabric surface was measured by an Attension Theta Flex Optical Contact Angle Meter (Biolin Scientific, Stockholm, Sweden) and the Laplace–Young model was chosen as the fitting method. The droplet volume was set at 5 μL and the WCA of the cotton fabrics was measured at 5 different locations of the fabric to achieve the average WCA value.

The micromorphology of the fiber surface was observed on a JEOL transmission electron microscope (TEM, 200CX, JEOL, Akishima, Tokyo, Japan). The cotton fibers before and after water repellent finishing were adhered to the conductive glue to spray gold. The surface appearance of the fiber was observed and the relationship between the surface micromorphology and water repellency was analyzed. The chemical composition of cotton fabrics before and after finishing was analyzed by X-ray photoelectron spectroscopy (XPS, Thermo Scientific K-Alpha, Waltham, MA, USA).

## 3. Results and Discussion

### 3.1. Preparation of Fluorine-Free Water Repellent Agent

In this study, CAHPS was first synthesized by an alkaline equilibrium polymerization reaction, using D4, APDES and HDTMS as raw materials. Then, CAHPS was reacted with modified nanosilica to prepare chemically bonded organic–inorganic nanohybrid cross-linked polysiloxane (rSiO_2_–CAHPS), which was then emulsified to obtain rSiO_2_–CAHPS water repellent (Scheme 1). It is a special finishing agent for cotton fiber. The side chain -C_16_H_33_ can provide it with good hydrophobicity. The hydroxyl groups can make it form a mild network cross chain structure and react with the hydroxyl groups of cotton fibers to improve the durability of the finishing effect. The polysilane main chain can form a layer of hydrophobic film, while silica makes it form a rough surface. Silica grafted chemically on the main chain can make it difficult to remove for a durable effect.

When synthesizing CAHPS, the amount of APDES is directly related to the ammonia value of CAHPS. As shown in Appendix A, the ammonia value increases linearly as the addition of APDES increases, thus the amount of -NH_2_ in CAHPS can be controlled by controlling the addition of APDES. An appropriate amount of APDES is beneficial for the subsequent reaction while too much -NH_2_ can cause yellowing of cotton fabric during the curing process. The content of the carbon chain silane coupling agent HDTMS has a great influence on the water repellency of CAHPS, as shown in Appendix A. With the increase in HDTMS content, the WCA of CAHPS film shows a trend of increasing first and then decreasing. This is because the more long alkane chain groups, the stronger the water repellency; however, excessive HDTMS affects the directional arrangement of CAHPS on the surface of the cotton fabric and results in a lower WCA.

The surface modification of nanosilica can improve the dispersion and effectively prevent the nanoparticles from agglomerating with each other. Moreover, the epoxy groups of GPTMS-SiO_2_ can react with the amino groups of CAHPS to increase the durability of the water repellency of finished fabrics through chemical bonding. When the silica of 100 nm was grafted onto the CAHPS molecular chain and GPTMS-SiO_2_ content was 15%, the WCA of cotton fabric finished with rSiO_2_–CAHPS could reach 141.7°.

The particle size distribution of nanoparticles in ethanol media before and after modification is shown in Appendix A. The average particle size of unmodified silica is 205.8 nm with a polydispersity index (PDI) of 0.274, indicating that the unmodified silica is easy to agglomerate. The particle size of the agglomerated structure reaches about 5000 nm. After being modified with GPTMS, the average particle size of GPTMS-SiO_2_ is 123.6 nm and the PDI is 0.056, demonstrating the homogeneity of GPTMS-SiO_2_. In other words, the presence of GPTMS can effectively prevent the agglomeration of nanoparticles and improve its dispersibility. The particle size of silica also affect the WCA of the films formed. rSiO_2_–CAHPS films prepared with 20 nm, 100 nm and 1 μm silica have WCA of 134.8°, 142.3° and 137.6°, respectively (Appendix A).

FT-IR spectroscopy is a powerful tool for analyzing and characterizing functional groups of polymers. After comparing the FT-IR spectra of CAHPS and GPTMS-SiO_2_ (Appendix A), we found that the symmetric and asymmetric absorption peaks of the C-H stretching vibration at 2936–2856 cm^−1^ become stronger, and the characteristic peaks of the Si-O-Si stretching vibration at 1097–1023 cm^−1^ are significantly wider and stronger due to the introduction of the GPTMS-SiO_2_. The disappearance of the absorption peak of the epoxy group at 902 cm^−1^ indicates that the ring-opening addition reaction proceeded smoothly [15], and the chemically bonded rSiO_2_–CAHPS target product was successfully prepared.

By comparing the TGA curves of rSiO_2_–CAHPS and CAHPS, it can be seen that the temperature of 5% weight loss of rSiO_2_–CAHPS is 190 °C and the weight loss temperature caused by thermal decomposition is concentrated around 400–600 °C (Appendix A). However, due to the covalent and hydrogen bonding between the GPTMS-SiO_2_ surface and CAHPS, and because the three-dimensional network inorganic crystal structure of silica restricts the thermal movement of the polymer chains, the thermal decomposition temperature of the internal chemical bonds of rSiO_2_–CAHPS is higher than that of CAHPS [38], indicating that rSiO_2_–CAHPS has better stability against high temperatures. In addition, the residual amount of rSiO_2_–CAHPS increases to 4.3% because of the introduction of GPTMS-SiO_2_. Therefore, the thermal stability of rSiO_2_–CAHPS is better than that of CAHPS, and a plausible explanation is that the covalent bond formed between the two components of CAHPS and GPTMS-SiO_2_ improves the heat resistance. The above thermal performance suggests that rSiO_2_-CAHP can be applied to the finishing of cotton fabric using a pad–dry–cure process.

### 3.2. Characterization and Analysis of Cotton Fabric Surface

#### 3.2.1. Surface Morphology of Cotton Fibers

As can be seen in Figure 1a, at low magnification (×3500), the surface of the unfinished cotton fibers is relatively smooth and the shape of a single fiber can be clearly seen with a diameter of approximately 10–20 μm. However, at high magnification (×10,000), there are many tiny grooves on the surface of the cotton fabric fibers. Figure 1b shows the microscopic appearance of the cotton fabric surface after finishing. It can be clearly observed that the surface of the cotton fiber is covered with a layer of film, the small grooves on the surface of the cotton fabric disappear, and at the same time randomly distributed nanoparticles can also be observed. At high magnification (×10,000), the microscopic morphology of the nanospheres is clearly visible, with no agglomeration between the nanoparticles and showing a good adsorption state. The film formed on the surface of cotton fibers is the cross-linked polysiloxane CAHPS, which has strong film-forming properties and is able to adhere to the surface of cotton fibers to form a thin and dense silicon film, reducing the surface energy of the cotton fabric substrate and improving the hydrophobicity of the silicon film as well as the binding fastness with the cotton fibers. The reason why the rough structure of the cotton fiber surface is significantly improved is the addition of nanosilica. These particles are tightly coated on the fiber surface to form a micro/nanostructure, which is also an important influencing factor in the preparation of a cotton fabric surface with excellent water repellency. Moreover, the uniformly distributed nanosilica has little effect on the air permeability of the cotton fabric due to its rough structure on the surface of the fiber, which acts as an ‘air cushion’ in the Cassie–Baxter model to a large extent [39].

#### 3.2.2. Elemental Analysis of the Cotton Fiber Surface

Figure 2a is the full spectrum scan of the original cotton fabric and the cotton fabric after finishing with rSiO_2_–CAHPS water repellent agent. The C ls peak (285.0 eV) and the O 1s peak (531.8 eV) are clearly visible on the curve of the unfinished cotton fabric, which is due to the fact that cotton fiber is mainly composed of C and O elements. The rSiO_2_–CAHPS water repellent film of the cotton fabric after finishing contains elements C, O, Si and N with elemental contents of 53.07%, 34.76%, 10.95% and 1.22%, respectively, so new peaks Si 2s (151.6 eV) and Si 2p (144.3 eV) are evident in the cotton fabric after finishing, but the N 1s (399.4 eV) peaks were not evident. This is because the addition of rSiO_2_–CAHPS was small and a very thin film with very little N element was formed, so no significant N 1s characteristic peak appears. Nevertheless, based on these characteristic peaks, rSiO_2_–CAHPS has been shown to form a thin film on the cotton fabric.

The high-resolution XPS spectra of the C 1s and Si 2p films on the surface of the cotton fabric after finishing are shown in Figure 2b,c. The presence of a faint C–N absorption peak near the high binding energy of 286 eV on the C 1s diagram is due to the low C–N content and possibly to the overlap of the C–O group with the absorption peak of the C–N group. In the Si 2p spectrum, the characteristic absorption peak of silica nanospheres at 103.5 eV is not evident, and only the characteristic absorption peak of Si–O–Si of CAHPS appears at the binding energy of 102.4 eV. It can be confirmed that rSiO_2_–CAHPS has been successfully coated on the surface of the cotton fabric and has undergone a directional arrangement. It is speculated that the nanoparticles and cotton fibers preferentially interacted with each other for adsorption, and were then coated under the polysiloxane, while the low surface energy CAHPS fraction was in the outer layer, forming a dense silicon film. This kind of arrangement is beneficial to reduce the surface energy of cotton fiber [40].

### 3.3. Performance Evaluation of Cotton Fabric after Finishing

#### 3.3.1. Water Repellency

As shown in Figure 3, the yellow drop on the finished cotton fabric shows a clear spherical shape, and the WCA is 149.5°. However, the yellow drop on the surface of the unfinished cotton fabric is completely absorbed, presenting good hydrophilic behavior. In Figure 3d, the finished cotton fabric floats without external forces, while the unfinished cotton fabric is moistened and sinks to the bottom of the water.

Cotton is a commonly used fabric. It can have better water repellency after finishing with rSiO_2_–CAHPS water repellents, resisting different conventional liquids in daily life. As shown in Figure 3, common liquid droplets, including salt water, tea, coffee, dying solution, milk and cola, are able to take a round ball shape and stay on the surface of the cotton fabric finished with rSiO_2_–CAHPS water repellent. Their corresponding WCA values are 147.8°, 148.5°, 148.3°, 150.2°, 145.9°, 148.6°, respectively, which clearly reveals that the cotton fabric finished with rSiO_2_–CAHPS water repellent has good stain resistance to common liquids.

In Figure 4a, the finished cotton fabric is shown immersed in water, and a shiny water film on the surface can be clearly observed because of the hydrophobic effect of the finished cotton fabric. The surface of its micro/nanorough structure will carry air into the fibers, and the surface is surrounded by bubbles, showing Cassie–Baxter non-wetting behavior, which results in a mirror-like effect on the surface of the finished cotton fabric [41,42]. Compared with the unfinished cotton fabric, it can be seen in Figure 4b that the cotton fabric is moistened rapidly in water due to the large number of hydroxyl groups on the surface. This proves once again that the water repellency of the cotton fabric is greatly improved after finishing with rSiO_2_–CAHPS water repellent.

#### 3.3.2. Air Permeability of Finished Cotton Fabric

The air permeability of the fabric plays an essential role in wearing comfort and breathable water repellent fabrics have a broad application prospect. However, the existing fabrics with excellent water repellent effect are often not breathable and provide a poor wearing experience.

As shown in Figure 5, we designed a simple experiment to test the air permeability of the fabric. The finished fabric was placed on top of a small bottle containing ammonia solution. When water droplets containing phenolphthalein reagent were dropped on the surface of the cotton fabric, the drops rapidly changed from colorless to pink, indicating that the cotton fabric finished with rSiO_2_–CAHPS water repellent still has good air permeability. The reason is that ammonia vapor diffuses spontaneously and the voids in the fabric are used as transport channels. After the cotton fabric is finished using the pad–dry–cure process, rSiO_2_–CAHPS reacts with the -OH of cotton fabric, and is firmly fixed to the individual fibers. Therefore, it hardly affects the air permeability of the cotton fabric and meets the daily wearing requirements of the textile.

#### 3.3.3. Serviceability of Finished Cotton Fabric

Textiles are often required to maintain their original properties after several washes in daily life, so the finished cotton fabric needs to have a certain washing resistance, which can also reflect the durability of the finishing effect of the functional finishing agent on the textile. Therefore, in this study, the effect of soaping times (5, 10, 15, 20, 25 and 30) as per the published paper by our colleague [43] on the hydrophobicity of the fabric was investigated.

As shown in Figure 6, the WCA of the finished fabric sample is still 144.2° after 15 rounds of soaping and 140.9° after 30 rounds of soaping, showing strong adhesion between the water repellent finishing agent and cotton fiber. This is because CAHPS is chemically bonded with GPTMS-SiO_2_, which is a special nanohybrid cross-linked polysiloxane. When cured at a high temperature, rSiO_2_–CAHPS undergoes an oriented arrangement on the fiber surface. Long carbon chain groups and methyl groups with low surface energy migrate towards the fabric surface and point towards the air, Si-O-Si dipole bonds point towards cotton fiber and nanosilica is encapsulated under the polysiloxane and point towards the cotton fiber matrix. The reactive Si-OH and C-OH of rSiO_2_–CAHPS will be effectively bonded with C-OH of cotton fibers, which can effectively improve the water resistance of cotton fabric after finished with rSiO_2_–CAHPS water repellent.

According to GB/T 3923.1-2013 standard and GB/T 17644-2008 standard for fabric mechanical property tests (Table 1), we found that compared with the original cotton fabric, the warp and weft breaking strength of cotton fabric finished with water repellent agent decreases, while the decline range is acceptable. After treatment, the warp strength is 322 N with a strength retention rate of 97.2%. The weft strength is 146 N with a strength retention rate of 96.7%. Overall, the change in strength is not significant and does not affect the serviceability of the cotton fabric. At the same time, the whiteness is slightly reduced, but it still can meet the basic application of cotton fabric.

The abrasion of fabric in use is one of the main types of damage to the fabric, so an abrasion resistance test is essential for the finished fabric, and it is an important index for assessing the fastness of the fabric. The wear resistance test of cotton fabric was conducted for cotton fabrics before and after finishing according to GB/T 3920-2008 standard and the results are shown in Figure 7. The water repellency of finished fabrics decreases slightly after 5 rounds of dry rubbing, with a WCA of 137.6°, which still maintains good water repellency. However, as the rounds of dry wiping continue to increase, the WCA also continues to decline. After 20 rounds of dry rubbing, the WCA of cotton fabric is only 112.2°, which is because after repeated dry rubbing, the surface structure of the cotton fabric is damaged, leading to the destruction of the silicon film formed on the surface of the fabric, and thereby reducing the WCA of the cotton fabric.

#### 3.3.4. Performance Comparison with Commercially Available Water Repellent

To compare water repellency, air permeability and serviceability with existing water repellents, the cotton fabrics were finished with rSiO_2_–CAHPS water repellent and commercial water repellent WP-107A according to the same finishing and testing conditions, including using the same washing process (Table 2). After finishing, there is little difference between them in WCA, air permeability and abrasion cycles (20 cycles). However, rSiO_2_–CAHPS water repellent still has a WCA as high as 142.8–143.4° after 20 standard washes, and even after 30 washing cycles, the WCA is 140.6–141.1°, while WP-107A could not be measured after 20 washing cycles due to quick absorption of the measured water by the cotton fabric, which means that the finishing agent was shed. This is mainly because the C-OH groups and Si-OH groups of rSiO_2_–CAHPS water repellent and the C-OH of cotton fibers are effectively bonded, leading to a stronger bond with the fabric and exhibiting excellent durability.

### 3.4. Proposal of the Film Formation Mechanism

The binding force between rSiO_2_–CAHPS water repellent and cotton fabric can be proved by the dry weight of the fabrics after a drying and steaming process before and after washing (Table 3). Therefore, 100 ± 0.005 g cotton fabric were taken to simulate the actual pad–dry–cure process (Scheme 2a), and the dry weight of the fabrics after drying and steaming stages is nearly the same before washing, but it is different after washing. The dry weight of fabric after drying but before washing is 105.992–106.008 g, but the dry weight after washing is 99.992–100.004 g, nearly equal to the weight of unfinished fabric. This is because after the drying stage, rSiO_2_–CAHPS water repellent is attached to the fabric with a weak force (physical adsorption), which is easy to wash out, while after the curing stage, the finishing agent is attached to the fabric with a stronger force (covalent bonding).

Based on weight analysis, SEM and XPS results, the film formation mechanism on the surface of the cotton fabric can be determined, as shown in Scheme 2b. When finishing the cotton fabric with rSiO_2_–CAHPS water repellent, the film formation process is mainly divided into the following three steps: firstly, through the dipping and nipping process, rSiO_2_–CAHPS water repellent is easily dispersed and attached to the surface of cotton fibers; secondly, during the drying step, the surface of the cotton fabric with rSiO_2_–CAHPS water repellent is dried at 100 °C, and then rSiO_2_–CAHPS film is closely arranged on the surface of cotton fiber; the final step is curing at high temperature. After curing, the rSiO_2_–CAHPS film undergoes a directional arrangement on the fiber surface, its long-chain alkyl groups with low surface energy migrate towards the surface, hydrophobic methyl and long-chain alkyl groups face outwards and nanoparticles are encapsulated by the polysiloxane and close to the cotton fiber matrix. In addition, the reactive C-OH groups and Si-OH groups of the side chain of rSiO_2_–CAHPS are covalently bonded with the C-OH groups of cotton fiber during high-temperature curing, which effectively improves the durability of water repellent on cotton fabric.

## 4. Conclusions

CAHPS was synthesized by an alkaline equilibrium polymerization reaction and silica was modified with GPTMS to prepare GPTMS-SiO_2_. rSiO_2_–CAHPS was then prepared by a ring-opening addition reaction, and a fluorine-free water repellent for cotton fabric was obtained after emulsification. When the amount of APDES added was 8%, HDTMS was 6%, 100 nm silica was selected for modification and the amount added was 15%, the WCA of rSiO_2_–CAHPS film applied to cotton fabric could reach 141.7°. Application test results show that the finished cotton fabric is resistant to conventional liquids and the fabric still has good water repellent properties after 30 soaping cycles. After finishing with rSiO_2_–CAHPS water repellent, the properties of cotton fabric are hardly affected, and the air permeability, breaking strength and whiteness of the fabric are not significantly reduced. SEM and XPS shows that the water repellent forms a dense silicon film on the surface of cotton fabric, and the presence of silica improves the nanorough structure of the fiber surface. The synergistic effect of cotton fiber matrix, nanosilica and CAHPS with low surface energy endows the surface of cotton fabric with a micro/nano-multi-scale microscopic rough structure, making the finished fabric exhibit an excellent water repellent effect. In addition, rSiO_2_–CAHPS water repellents are bonded to cotton fibers though active groups, which can maintain a long-lasting water repellent effect for cotton fabric after finishing. In conclusion, the finishing agent developed in this study is a fluorine-free water repellent with excellent and durable water repellent properties.

## Data Availability

The data that support the findings of this study are available from the corresponding author, Yu Liu, upon reasonable request.

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
