# Peer review of "Preparation and Application of Fluorine-Free Finishing Agent with Excellent Water Repellency for Cotton Fabric"

_polymers, 2021, doi:10.3390/polym13172980_

Round 1

Reviewer 1 Report

The authors are proving the superiority of the developed fluorine-free finishing agent. The author's efforts, such as various analysis results and performance tests, are worthy of evaluation. However, the purpose of this study and the authors' intention is unclear, so the following major revision is necessary.

Major comments

  1. The characterisation of the finishing agent and the surface treated cotton is divided into several places, which hinders understanding. After characterisation (WCA, TEM, XPS), it would be good to organise them in the order of performance evaluation. In addition, IR or DSC analysis is not considered necessary in the main discussion part.

  1. Comparison with existing finishing agents with Flourorine is insufficient. Only those with and without the developed finishing agent are compared. In the case of commercial finishes, it is briefly mentioned in the description of Figure 6, but one of the methods in Figures 3, 4, 5 needs comparison. Also, the data needs to show that WP-107A has no water repellent at 20 abrasion cycles.

  1. Regarding Scheme 2, the author explains the film formation with SEM and XPS results, but it is difficult to relate to which part this speculation is. At the very least, there should be a comparison of data for each drying process.

Minor comments

  1. It is called New type of CAHPS. Does this mean that the author developed it for the first time?
  2. For ring-opening addition, also indicate a reference if it is not the original method.
  3. Line 94: Write the original name of CAHPS.
  4. Line 125: Write the material referred to by the catalyst.
  5. Line 204: Write the original name of the WCA.
  6. Line 223: How the size different SiO2 samples were prepared?
  7. Figure 4(a): Is there any comparison with uncoated or commercial ones?
  8. Figure 4(b)(c): Is it related to breathability?
  9. Figure 5: What is the criterion for good resistance? Does it mean WCA 140Ëš or higher?
  10. Figure 5: As shown in Fig. 6, it usually decreases as a logarithmic or exponential function, but what is the reason why WCA decreases linearly in this sample?

Author Response

Dear Editors and Reviewers:

We thank you very much for giving us an opportunity to revise our manuscript. The comments and recommendations were helpful, and we greatly appreciate them. The revised portions are marked in yellow in the paper according to the reviewers’ comments.

The main corrections in the paper and the responses to the comments are as follows:

Reviewer #1:

Major comments

1.The characterisation of the finishing agent and the surface treated cotton is divided into several places, which hinders understanding. After characterisation (WCA, TEM, XPS), it would be good to organise them in the order of performance evaluation. In addition, IR or DSC analysis is not considered necessary in the main discussion part.

Response: Thanks for your work and comments. We rearranged the content of ‘Results and Discussion’ in the paper, we wrote the characterization (XPS, TEM) first, and then the performance discussion. In addition, we move IR and DSC to the supplement and simply wrote some important conclusions in the paper.

2.Comparison with existing finishing agents with Flourorine is insufficient. Only those with and without the developed finishing agent are compared. In the case of commercial finishes, it is briefly mentioned in the description of Figure 6, but one of the methods in Figures 3, 4, 5 needs comparison. Also, the data needs to show that WP-107A has no water repellent at 20 abrasion cycles.  

Response: The existing finishing agents with fluorine has excellent water repellency. Due to the reasons of molecular structure, its water repellency can not be achieved by fluorine free water repellent agent, and can only be as close as possible. Finishing agents with fluorine often contain perfluorooctane sulphonate (PFOS) and perfluorooctanoic acid (PFOA), which are difficult to degrade and have high toxicity and bioaccumulation. For these reasons they have been restricted to use. Furthermore, all finishing agents with fluorine will be restricted to use in dyeing industry in the near future, thus the development of fluorine free water repellents is of great importance.

WP-017A is a commercially available water repellent with superior performance, but it is only adsorbed on the fiber surface by van der Waals force. Therefore, in addition to its washing resistance, the other performance including WCA after abrasion cycles, air permeability and water repellency of cotton fabric after finishing should be nearly the same as rSiO2-CAHPS water repellent. But washing resistance is always thought to be one of the most important performance in clothing. For this consideration, we think originally that it is not necessary to be compared in Figure 3, 4, 6.

We agree with the reviewer, comparison with existing finishing agents is meaningful. We compared the performance of finishing agents with commercially available water repellent WP-107A. We revised and added some result and discussions for WP-017A to compare with rSiO2-CAHPS water repellent in the paper. Because it is difficult to obtain fluorine-containing water repellent from cooperative enterprises, we only used WP-107A for a performance comparison in the paper.

 3.Regarding Scheme 2, the author explains the film formation with SEM and XPS results, but it is difficult to relate to which part this speculation is. At the very least, there should be a comparison of data for each drying process.

Response: It is a good idea. We added a comparison table of weight data of cotton fabrics after drying and steaming process with and without washing.

Minor comments

1.It is called New type of CAHPS. Does this mean that the author developed it for the first time?

Response: Thanks for your question. We revised the expression, CAHPS is developed for the first time.

2.For ring-opening addition, also indicate a reference if it is not the original method.

Response: Thanks for your suggestion. We added a reference.

3.Line 94: Write the original name of CAHPS.

Response: Thank you for reminding us. We added the original name of CAHPS. 

4.Line 125: Write the material referred to by the catalyst.

Response: We revised the catalyst with its material name.

5.Line 204: Write the original name of the WCA.

Response: We wrote the original name of the WCA in the first place of the paper, then wrote the abbreviation ‘WCA’.

6.Line 223: How the size different SiO2 samples were prepared?

Response: We added the resource of SiO2 with different size in 2.1 Material.

7.Figure 4(a): Is there any comparison with uncoated or commercial ones?

Response: They are actually the same as rSiO2-CAHPS water repellent. We revise the result and discussion and added comparison in a table and discussed.

8.Figure 4(b)(c): Is it related to breathability?

Response: This is indeed improper and easily misleading. We moved these figures and discussion from Figure 4 to be a separate Figure, and use ‘air permeability’ instead of ‘breathability’ for clear expression.

9.Figure 5: What is the criterion for good resistance? Does it mean WCA 140Ëš or higher?

Response: Thanks for your comment. We revised the expression in detailed data, deleted ‘good resistance’.

10.Figure 5: As shown in Fig. 6, it usually decreases as a logarithmic or exponential function, but what is the reason why WCA decreases linearly in this sample?

Response: Thanks you for pointing out the abnormal data. We checked and measured these data, and redrew Figure 5 and 6 in our Manuscript.

Reviewer 2 Report

The manuscript entitled is a valid experimental work devoted to the design, investigation and proof-of concept of a fluorine-free water repellent targeted to excellent and durable water repellent properties. The topic of the manuscript fits well the scope of the journal. The introduction section clearly describes the state-of-the-art in the specific field of research. The presented results are appropriately and carefully described. The obtained results might have definite practical impact. The conclusions are clearly substantiated by the gained results. The overall quality of the manuscript is high. With pleasure I can recommend the manuscript for acceptance in the present form. Minor revision of the style of the language might be necessary; this will increase the overall quality of the work.   

Author Response

Reviewer #2: The manuscript entitled is a valid experimental work devoted to the design, investigation and proof-of concept of a fluorine-free water repellent targeted to excellent and durable water repellent properties. The topic of the manuscript fits well the scope of the journal. The introduction section clearly describes the state-of-the-art in the specific field of research. The presented results are appropriately and carefully described. The obtained results might have definite practical impact. The conclusions are clearly substantiated by the gained results. The overall quality of the manuscript is high. With pleasure I can recommend the manuscript for acceptance in the present form. Minor revision of the style of the language might be necessary; this will increase the overall quality of the work.   

Response: Many thanks for your kind comments. In this manuscript, based on the structural characteristics of cotton fiber, we synthesized an novel organic-inorganic nano-hybrid cross-linked polysiloxane, and then emulsified to get fluorine-free water repellent. The water repellent cotton fabric can be carried out only using the pad-dry-cure process, instead of special equipment and harsh preparation conditions. The finished cotton fabric has excellent water repellency, air permeability, serviceability, and it has long-term finishing effect.

Reviewer 3 Report

I would recommend including into introduction part discussion of the paper: "Fabricating durable, fluoride-free, water repellency cotton fabrics with CPDMS, Journal of Applied Polymer Science 135(25):46396, DOI: 10.1002/app.46396

Other remarks, mostly editorial:

Line 43: Although the organic fluorine water repellents containing eight carbon

 fluoride feature excellent water repellent properties REMARK: what do you mean by “containing eight carbon fluoride” ?

Line 153: then stirring violently for 15 min BETTER followed by violent stirring  for 15 min

LIne 243: It also means that the hydroxyl groups on the surface of nano-silica have

reacted with the coupling agent GPTMS and successfully grafted on the surface of nano-silica, rather than a simply physical blend with silica BETTER It also means that the hydroxyl groups on the surface of nano-silica have reacted with the coupling agent GPTMS, which was successfully grafted on the surface of nano-silica, rather than forming a simple physical blend with silica

Line 253  DSC analysis CORRECT: TGA Analysis

Author Response

Dear Editors and Reviewers:

We thank you very much for giving us an opportunity to revise our manuscript. The comments and recommendations were helpful, and we greatly appreciate them. The revised portions are marked in yellow in the paper according to the reviewers’ comments.

The main corrections in the paper and the responses to the comments are as follows:

Reviewer #3: I would recommend including into introduction part discussion of the paper: "Fabricating durable, fluoride-free, water repellency cotton fabrics with CPDMS, Journal of Applied Polymer Science 135(25):46396, DOI: 10.1002/app.46396

Response: Thank for your recommendation. It is an excellent paper about fluoride-free water repellency cotton fabrics. We added this reference.

Other remarks, mostly editorial:

Line 43: Although the organic fluorine water repellents containing eight carbon  fluoride feature excellent water repellent properties REMARK: what do you mean by “containing eight carbon fluoride” ?

 Response: Thank you for pointing out our misunderstanding. We rewrote the sentence.

Line 153: then stirring violently for 15 min BETTER followed by violent stirring  for 15 min

 Response: Thank for your correction. We corrected the sentence.

LIne 243: It also means that the hydroxyl groups on the surface of nano-silica have reacted with the coupling agent GPTMS and successfully grafted on the surface of nano-silica, rather than a simply physical blend with silica BETTER It also means that the hydroxyl groups on the surface of nano-silica have reacted with the coupling agent GPTMS, which was successfully grafted on the surface of nano-silica, rather than forming a simple physical blend with silica

 Response: Thank for your suggestion. We revised the sentence.

Line 253  DSC analysis CORRECT: TGA Analysis

 Response: Thank for your correction. We corrected the sentence.

Round 2

Reviewer 1 Report

The authors have made revisions based on a sufficient understanding of each comment, so this paper is worthy of publication. However, since the following contents are considered to require a little explanation, an additional revision is requested.

1) Table 2, It should be WP-107A not WP-101A? and washing cycle data is miswritten. Please double-check the values.

2) It is still difficult to understand the film formation mechanism. first, what is after drying and what is after steaming? and why the weight incrased after drying with not washing process?? and, does the weight should be incrased after film formation?

Please give more detail description of method and intend of this experiment for general readers.

Author Response

1) Table 2, It should be WP-107A not WP-101A? and washing cycle data is miswritten. Please double-check the values.

Response: Thanks for your correction. We corrected Table 2 about WP-107A. After checked, for commercial WP-107A water repellent, WCA could not be measured after 20 times washing due to quick absorption of the measured water by the cotton fabric, while the data of rSiO2-CAHPS water repellent is 30 washing cycles. In order to avoid readers' misunderstanding, we also added 20 washing cycles in the meantime. Furthermore, we correct the wrong ‘water repellant’ with ‘Water repellent’ in some expressions.

2) It is still difficult to understand the film formation mechanism. first, what is after drying and what is after steaming? and why the weight incrased after drying with not washing process?? and, does the weight should be incrased after film formation?

Response: Thanks you for your reminder. For a reader who is not a dyeing major, it is really difficult to understand. We added the finishing flow in Scheme 2 for more intuitive understanding of the pad-dry-cure process. At the same time, we also made some necessary explanations, such as ‘This is because after the drying stage, rSiO2-CAHPS water repellent is attached to the fabric with weak force (physical adsorption), which is easy to wash out, while after the curing stage, the finishing agent is attached to the fabric with stronger force (covalent bonding).’

Round 3

Reviewer 1 Report

The paper is well revised and significantly improved.